# Navigating pathways to develop self-compassion in sport, dance, and music: Qualitative insights from volunteer participants on balancing criticism and compassion, the pivotal influence of the social environment and strategies to overcome setbacks

Céline Kosirnik[1]*, Roberta Antonini Philippe[1], Valentino Pomini[2]

1 Institute of Sports Science, Faculty of Social and Political Science, University of Lausanne, Lausanne, Switzerland, 2 Institute of Psychology, Faculty of Social and Political Science, University of Lausanne, Lausanne, Switzerland

* celine.kosirnik@unil.ch

## Abstract

Self-compassion has emerged as a valuable skill for performers, contributing to both their well-being and performance enhancement. However, the question remains: how can it be effectively integrated and cultivated within performers' daily practice? This study examines the extent to which self-compassion is integrated by performers, its practical application, and the role of the social environment in shaping these skills. A total of 27 performers (athletes, dancers, and musicians), aged 14–27, practicing at a pre-professional level, participated in semi-structured interviews. These interviews explored performers' responses to challenging situations during practice, the coping strategies employed, the use or non-use of self-compassion, and the influence of their social environment. Through a thematic analysis, six key themes emerged, each comprising subthemes: Explore your inner critic; Balance over-identification; Use strategies to overcome setbacks; Create a virtuous circle; Acknowledge your social environment; and, Learn about yourself. The findings revealed that performers continue to place significant emphasis on self-criticism, often neglecting self-kindness when most needed. Despite this, they recognize the benefits of self-compassion and report various strategies to manage difficulties. Additionally, the importance of the coach's communication style and the performers' desire to be better understood by their coach were emphasized. This qualitative study provides deeper insights into the experiences of performers and offers practical guidance on how self-compassion can be more effectively integrated into their daily practice.

**Data availability statement:** All relevant data are within the manuscript and its Supporting information files.

**Funding:** The author(s) received no specific funding for this work.

**Competing interests:** The authors have declared that no competing interests exist.

## Introduction

"My movement has to be perfect", "I'm not allowed to make any mistakes", or "I absolutely mustn't mess up"—these are the frequent inner dialogues of performers. Sports, music and dance are competitive worlds in which there is little room for error or failure in one's performance, as the ultimate goal is to be the best. Sports settings often emphasize mental toughness: "Be more consistent and better than your opponents in remaining determined, focused, confident, and in control under pressure." ([1] p209). This mindset is essential in performance activities; however, it does not encourage performers to look at themselves with kindness and understanding—instead, they challenge themselves constantly. Recent research highlights the value of combining self-compassion and mental toughness to enhance both performance and wellbeing [2,3], particularly in sports, with emerging evidence in music and dance [4], although research in these latter fields is still scarce.

### Self-compassion

Self-compassion is defined by three components [5]: self-kindness, common humanity and mindfulness. Self-kindness refers to the ability to grant oneself the care and kindness needed when experiencing a difficult situation. Common humanity represents the idea that human beings are not alone in their experiences, and it aims for interconnection between individuals. Finally, mindfulness refers to the ability not to get caught up in a situation or, more specifically, in the emotions or thoughts it might generate but to remain in the present moment without passing judgment on what is happening to oneself. As [5] stated, developing self-compassion is developing a "healthy self-attitude" towards oneself, especially during moments of need.

### Self-compassion and the world of performance

Researchers have explored different sports to better understand the concept and specificities of self-compassion and how it applies to performance and wellbeing. [6] studied women athletes in Canada and found that self-compassion can have a lasting and adaptive impact on some dimensions of psychological flourishing in sport (i.e., purpose in life and self-acceptance). Also, [7] investigated the link between perceived sport performance, self-compassion and self-criticism. They found that "self-compassion is related to higher perceived sport performance, whereas self-criticism is unrelated to perceived sport performance" (7 p304). [8] discovered interesting results about self-compassion and the perception of athlete's performances. It seems that self-compassion is associated with a better subjective evaluation of one's sporting performance, but only in the days following mediocre performance. Finally, a recent literature review [9] highlighted self-compassion's varied contributions and results in the world of sport. [9] showed that there was growing interest in self-compassion in sport but that it remained on the sidelines in the performing arts (e.g., music or dance).

Developing self-compassion is linked to an increase in positive psychological factors, such as self-esteem [10], self-acceptance [11], or optimism [12] and flow [13].

Self-compassion seems to be particularly useful for overcoming setbacks and reducing levels of depression, anxiety and stress [14,15]. Researchers have also shown that athletes with higher levels of self-compassion perform better and deal more effectively with stress factors [16] or anxiety [17,18].

Self-compassion is still a new area of research in the performing arts (e.g., dance or music) [4]. [19] pointed out that a dancer who had just made a mistake could show empathy for himself by treating himself as he would a friend instead of accepting intrusive thoughts of harsh self-criticism. The deleterious impact of dancers' practice environments has been contrasted with the protective effects of developing self-compassion [19,20]. [21] also highlighted the deleterious consequences of constant evaluation in the mirror, whether by peers or oneself, which is, unfortunately, not conducive to self-compassion and kindness, leading to harmful effects on one's self-image. In the world of music, researchers have highlighted a link between levels of performance anxiety, stress and levels of self-compassion [22], particularly over-identification with a performance situation and how developing self-compassion can help musicians take a step back from a specific situation [23]. [24] also showed that self-kindness has a positive influence on music performance anxiety thus suggesting that developing positive talk could help musicians during their performances. Furthermore, [25] emphasized "positive connections between coping, resilience, self-compassion, health, and wellbeing among conservatoire music students" (25 p4). Finally, [26] studied the impact of self-compassion intervention on college music students and highlighted the positive influence of self-compassion on their mental health. Self-compassion seems, therefore, to give rise to proactive and more adaptive behaviours [27], increasing one's well-being and improving one's overall performance [28].

Nevertheless, this concept still sometimes frightens performers striving for excellence and who are frequently convinced that self-criticism is necessary for optimal psychological functioning [28], to not slack off [29] and to perform at an elite level [30]. They fear that self-compassion will lead to self-pity or that they will become too indulgent with themselves, leading to passivity [29,31]. A variety of authors has provided evidence that self-compassion is more beneficial than self-criticism in terms of the ability to adapt [20,30] or bounce back from failure [32].

## Self-compassion and the influence of one's social environment

The influence and role of athletes' social environments have also been of interest in the development of self-compassion skills [33]. [34] noted that coaches aim to show kindness rather than explicitly use compassion, but the performance environment's focus on perfection and selection limits its application. Furthermore, [35] suggest that more supportive and compassionate coaching enhances athletes' well-being and performance. Researchers highlighted that positive, supportive [36], compassionate [37] relationships can help to strengthen self-compassion [38]. Role models such as coaches appear to play a key role in developing a caring attitude toward oneself [39]. Furthermore, it seems that the more compassion a person perceives in their peers, the more likely they will develop self-compassion [40]. In some environments, self-compassion may be seen as a weakness [31]; in others, it may be valued and encouraged [41]. Environments that promote self-acceptance and compassion are more conducive to the development of self-compassion [41,42]. [41] explain that coaches who value self-compassion through open communication and by being encouraging towards their athletes, help athletes to take a step back and recover more quickly from difficult sporting situations. Furthermore, it seems that increasing one's level of self-compassion also benefits one's environment as it enables more positive social comparisons [37,43] and increases one's ability to seek help from others [44].

[9] raised some interesting points for further research. Quantitative studies have predominated to date, and they underlined the issue of studying self-compassion as a domain-general or domain-specific concept. The present study aimed to partly respond to these recommendations as it is a qualitative research study and collected data from three different populations: dancers, musicians and athletes. The study's goals were to deepen our knowledge of self-compassion in these populations and how they use, or not, self-compassion skills when facing difficulties. Also, another aim is to examine the influence of their respective practice environments on their levels of self-compassion. Given the variety within these

populations, the term *performer* will be used to refer to all types of athletes, dancers and musicians, and the term *coach* will be used to refer to the coaches and, music and dance teachers.

## Materials and methods

### Participants

We enrolled 27 performers, from Switzerland's French-speaking region, into this study (22 women and five men), including eight dancers (ballet, modern or contemporary dance), eight musicians (playing wind, string and percussion instruments) and 11 athletes from gymnastics, acrobatic rock and roll, diving, judo, volleyball and badminton. Twenty of the performers practised their activity at a national or international level, spending a mean of 18.36 hours per week training (SD = 7.6), with a minimum of 10 and a maximum of 35 hours (Table 1). Study inclusion criteria were (a) being at least 14 years old, (b) practising for at least 10 hours a week and (c) being able to read and speak French.

### Study design

Participants took part in a one-on-one, qualitative semi-structured interview [45], lasting an average of 60 minutes, following an interview guide (S1 Table). These semi-structured interviews facilitated an in-depth exploration of the concept of self-compassion through a partially guided yet flexible format [45]. The recruitment and interviewing process took place between November 16, 2020 and August 25, 2022. Participants were recruited in three different ways: through dance schools, music schools and sports clubs; through direct contacts with coaches via LinkedIn; and, finally, through snowball recruitment based on referrals from our initial interviewees. Prior to their interviews, participants provided written informed consent. Minors participating in this study were permitted to sign the consent form independently, as per guidelines from SwissEthics, which entails that research with minimal risks does not require the informed consent of the legal representatives (parents) (S1 Checklist). Furthermore, the study received ethical approval from the Research Ethics Committee of the University of Lausanne (E_SSP_092020_00002 and E_SSP_082023_00002). All the personal data collected were coded and anonymized before analysis. Interviews were conducted in French by the first author, and the verbatim transcriptions presented in this article were then translated into English. The interviews took place either in person or online via Zoom for the convenience of some participants.

### Semi-structured interviews

The interview guide (S1 Table) was constructed using and inspired by [46] recommendations about differentiating the process with regard to a specific theme—in our case, self-compassion—and the importance of the questions putting participants at ease. One exploratory pilot interview was conducted with an expert from each discipline (i.e., dance, music and sport) to test and discuss the interview guide (S1 Table) and ensure that the questions and language used was appropriate and comprehensive. The core of the interview focused on the performers' personal experiences of a recent difficult situation and how they reacted to it. The participant was asked to describe the situation in detail before looking into their reactions

**Table 1. Participants information by discipline.**

| | N | Age (Mean ± SD) | Sex | Start of activity (age in years) | Start of an intensive training programme (age in years) | Performance level (N) | Hours of training per week |
|---|---|---|---|---|---|---|---|
| **Dance** | 8 | 17.4 ± 2.0 | F = 8 M = 0 | 6.4 ± 3.1 | 12.5 ± 2.1 | National = 4 International = 4 | 20.6 ± 8.1 |
| **Music** | 8 | 22.4 ± 3.7 | F = 6 M = 2 | 7.3 ± 1.8 | 15.7 ± 1.0 | National = 3 International = 5 | 21.0 ± 9.4 |
| **Sports** | 11 | 17.2 ± 1.9 | F = 8 M = 3 | 6.6 ± 2.2 | 12.1 ± 2.5 | Regional = 2 National = 3 International = 6 | 15.3 ± 5.6 |

When the format "6.6 ± 2.2" is used, it presents the mean ± the standard deviation.

and behaviours during that situation. They were then asked to react to three sentences representing the three components of self-compassion (S2 Table). Then, participants were questioned about their social environment (coaches, peers, parents) and its role in their practice. The focus was on coaches and coach–performer interactions during training or practice. As a closing question, participants were asked about what they would need to further develop their self-compassion.

## Data analysis

We performed a mixed inductive–deductive analysis, also called hybrid analysis [47]. Combining deductive and inductive analysis enhances transparency and supports the application of theory, strengthening the study's trustworthiness and relevance [48]. Some codes were predefined by the theoretical framework of self-compassion (i.e., self-kindness versus self-judgment, common humanity versus isolation, and mindfulness versus over-identification), while others emerged from the data. The first author read each interview several times to get familiar with the material. Data were coded using Nvivo software. Coding was inspired by [49] and [50] familiarisation and coding theme development, refinement and coding processes. Pseudonyms were used to guarantee the participants' anonymity.

## Results

### Developing self-compassion

The interview analysis identified six primary themes, each comprising different sub-themes (Fig 1). The results provide a deeper understanding of how performers incorporate or fail to incorporate self-compassion into their training or practice. The findings also provide insights on strategies that can enhance the development of self-compassion and highlight how social factors influence that development (Table 2).

### Explore your inner critic

**Your inner critic as a barrier.**  Most of the performers described themselves as being "demanding" (Ana, Ballet, 20), "serious" (Lucy, Ballet, 14) and having a very "rigorous work ethic" (Jennifer, Tuba, 23), although they were sometimes also "insecure" (Rose, Rhythmic gymnastics, 17). In addition, one unifying character trait among all the performers, whether they came from the world of music, dance or sports, was their tendency to seek perfection. They exhibited a mentality that left no room for error: "I want it to be perfect, so if I feel there's something wrong, I'll want to redo it." (Stephanie, Trampoline, 21).

The vast majority of performers explained that when they encountered a difficulty, whatever their mood, they found it difficult to distance themselves from the event and surrendered to their negative inner criticism: "Frankly, there are a lot of insults [uttered] when I'm on the board or even when I've finished a dive." (Ian, Diving, 17) Many performers said that they felt "rubbish" (Jennifer, Tuba, 23) if they had not achieved their goals of perfection and success in training or, even more so, in competition.

> "I know that sometimes I blame myself, for example, 'What you're doing isn't right.' And before a competition, it happens more and more. And then at the competition, 'Damn! The last two weeks were a disaster. How are you going to manage on stage now?'" (Linda, Acrobatic rock and roll, 19)

Excessive criticism seems to have its limits and performers realize it:

> Interviewer: "Before a dive, when you put yourself down, do you feel it helps you?"
>
> Ian: "Up until now, it's never really worked, but I can't really do it any other way, and I think I've already tried to do the opposite, to say to myself, 'Okay. No, it's okay. I'm here because I'm good.' And that's never really changed." (Ian, Diving, 17)

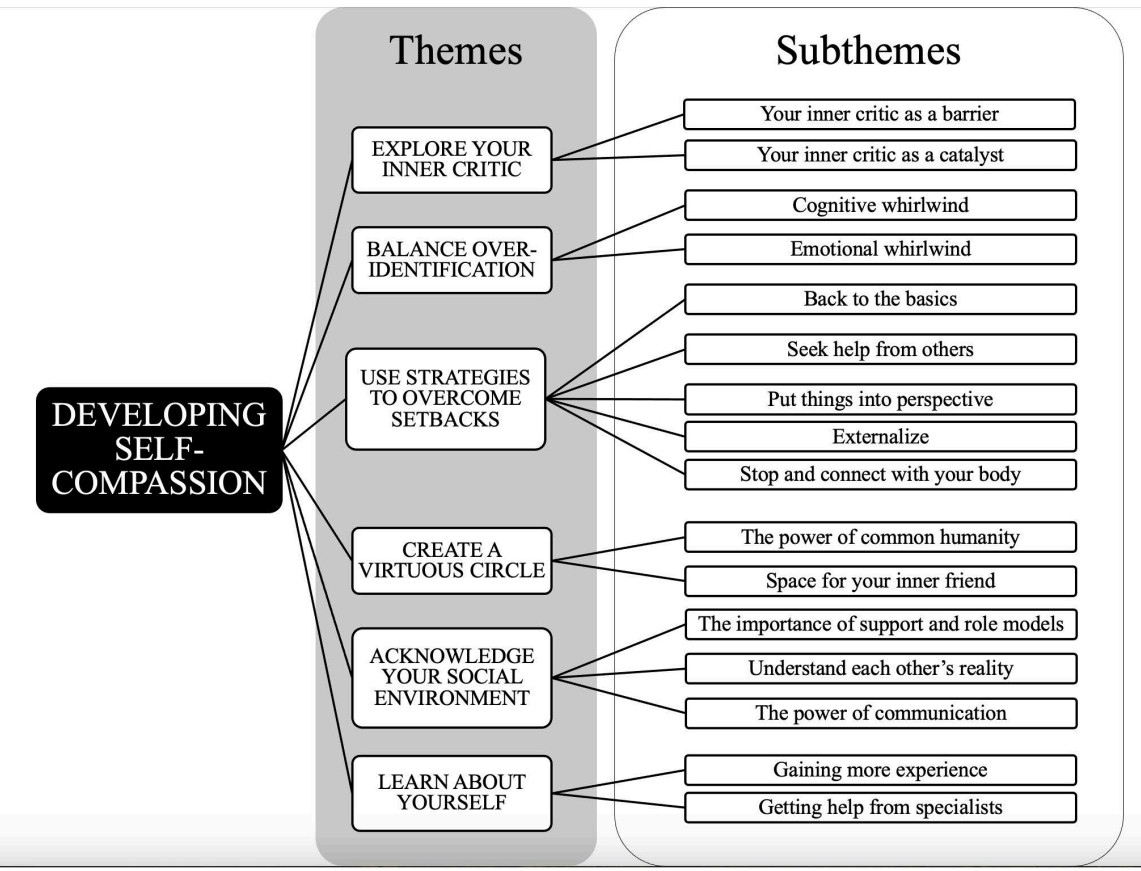

**Fig 1. Developing self-compassion: themes and subthemes.**

**Your inner critic as a catalyst.** Most of the performers seemed to suggest that they needed this self-criticism to perform better: "I'm not very satisfied with my work in the sense that I'm always looking to improve. No matter how good or bad the result is, I'm always looking for mistakes, so I'm not often proud of myself." (Ian, Diving, 17). Moreover, performers sometimes seemed to suggest that self-criticism lead to rigour and discipline: "I'm being unkind to myself, but that's what's going to make me train more rigorously and do better." (Linda, Acrobatic rock and roll, 19).

The performers never seemed to be satisfied. They expressed their need for this dissatisfaction as a booster. It "pushes them to be better and to just want perfection" (Emmy, Ballet, 17).and make them improve.

"In fact, in dance, you always have to question yourself and the fact that you can see your faults, that you can see what's not right, helps you to say, 'That's what I need to work on.' It's still a quality to be able to do that, but afterwards, it's important not to let it take over." (Elena, Ballet, 17)

### Balance over-identification

**Cognitive whirlwind.** The performers' comments highlighted the difficulties they encountered in managing their inner speech during difficult situations or when they made mistakes: "I'm aware that I'm speaking badly to myself, but I can't stop [...] And I can't say to myself, 'Hang on, you're speaking badly to yourself now. You need to calm down." (Ian, Diving,

**Table 2. Results' overview.**

| Themes | Subthemes | Results overview |
|---|---|---|
| EXPLORE YOUR INNER CRITIC | Your inner critic as a barrier | The pursuit of perfection fueled a harsh inner critic that, despite aiming to boost performance, often harmed performers' confidence and well-being. |
| | Your inner critic as a catalyst | For performers, self-criticism was seen as a driver of discipline and improvement, as long as it remained controlled. |
| BALANCE OVER-IDENTIFICATION | Cognitive whirlwind | The performers struggled to manage negative inner dialogue, often dismissing positive feedback, making it difficult to break the cycle of self-criticism. |
| | Emotional whirlwind | Performers often felt overwhelmed by emotions and tension, struggling to break the cycle of frustration and self-criticism. |
| USE STRATEGIES TO SETBACKS | Back to the basics | To cope with difficulty, performers returned to basic tasks, using repetition to regain control—though this strategy was not always effective. |
| | Seek help from others | Performers sought help through advice, observation, and positive feedback from others to overcome challenges. |
| | Put things into perspective | Performers used experience, positive reflection, and support to put setbacks into perspective. |
| | Externalize | Expressing emotions helped some performers cope with difficulties, though it sometimes reflected ongoing struggles rather than resolution. |
| | Stop and connect with your body | Performers coped with difficulties by pausing to rest, reconnect with their bodies, and mentally rehearse to regain focus. |
| CREATE A VIRTUOUS CIRCLE | The power of common humanity | Performers found comfort and motivation in shared experiences and peer support, though competition sometimes led to feelings of isolation. |
| | Space for your inner friend | Despite frequent self-criticism, some performers tried to practice self-compassion through positive self-talk, though often tempered by doubts about their own abilities and achievements. |
| ACKNOWLEDGE YOUR SOCIAL ENVIRONMENT | The importance of support and role models | Performers relied on parental support and viewed coaches as trusted role models whose guidance deeply influenced their development and confidence. |
| | Understand each other's reality | Performers sought coaches who demonstrate empathy and understanding of their unique experiences to deliver personalized and effective support. |
| | The power of communication | Performers valued clear, balanced coach communication—combining constructive feedback and positive reinforcement—to enhance motivation, trust, and perceived support. |
| LEARN ABOUT YOURSELF | Gaining more experience | Performers believe that gaining experience—through personal challenges, peer exchanges and coach guidance—is essential for developing self-compassion. |
| | Getting help from specialists | Performers emphasize the importance of specialized support mental health professionals to gain objectivity and develop self-compassion during challenging times. |

17). It seems almost impossible for some of them not to be hard on themselves. "[They're] aware that [they're] being hard on [themselves], but [they] just can't help it" (Emmy, Ballet, 17). Distancing from their negative cognitions seemed even more difficult when the coach was in charge: "I didn't have any decision-making power, so if the coach said it was rubbish, then it was rubbish." (Stephanie, Trampoline, 21).

It was also striking that even when the performers received positive feedback, they did not hear it. It was as if positive feedback did not lead to progress and improvement: "Then every time someone gives me feedback, I say to myself, 'Yes, but he's just saying that because he wants to be nice. He's not telling the truth about what he's heard and what he's analysed." (Cloe, Trumpet, 22)

"So, yes, generally, if I miss something, it cancels out a lot of good things very quickly because it annoys me to let something go and not be able to show what I can do." (Olivia, Rhythmic gymnastics, 15)

Achieving success or perfection sometimes seemed to be the only thing that allowed the performers to be proud of what they were doing.

"Generally speaking, I prefer to be told it's no good rather than being told it's very good—because it's never good enough." (Ana, Ballet, 20)

**Emotional whirlwind.** In difficult situations or when they had difficulty carrying out their task, performers mainly felt "frustration" (Laura, Ballet and contemporary dance, 17), "anger" (Kate, Contemporary dance, 20), "guilt" (Cloe, Trumpet, 22), "sadness" (Julia, Rhythmic gymnastics, 15) or "shame" (Victoria, Flute, 27). They seemed to have difficulty distancing themselves from their immediate emotions.

"But, in fact, it's more like I'm thinking 'This can't be happening. I mean, what am I doing?', I get angry and say'What the hell are you doing?" (Felix, Badminton, 18)

The vicious circle of emotions that get in the way during a difficult situation seemed to be accompanied by physical reactions that expressed performers' discomfort at the time: "Tension in the shoulders, in the lower back, breathing a bit faster," (Cloe, Trumpet, 22) or even "crying" (Olivia, Rhythmic gymnastics, 15).

Performers who expressed the most difficulty distancing themselves from a difficult situation, together with the emotions and thoughts associated with it, explained that certain situations helped them avoid falling into this vicious circle.

"I get angry, and I can't do anything. I try to stay calm, too, so I can set a good example for the younger ones, and I try to concentrate on myself and not look at other people, so I don't lose myself in other things. I try to concentrate and not get too angry with myself." (Rose, Rhythmic gymnastics, 17)

### Use strategies to overcome setbacks

Several strategies seem to have been used by performers, some more effective than others, to overcome the difficult situations during which they entered a vicious circle that hindered their performance.

**Back to the basics.** In the event of repeated difficulties or errors, performers tried to "play slowly, one hand after the other" (Maria, Piano, 26), "work well on a short passage" (Jennifer, Tuba, 23) or "do easy things that [they] get right [to] put [them] back in a good mood and calm them down" (Olivia, Rhythmic gymnastics, 15). The performers overcame difficult situations by concentrating on simple, basic movements or one they could always do well: "First of all, it's more complicated to be precise when you're upset. So, doing easy things that I manage to do will put me back in a good mood and calm me down, so I won't be shaking." (Olivia, Rhythmic gymnastics, 15). They also explained that they had to repeat the same thing several times for it to work: "I repeat it, I repeat it, and it works, and that's it." (Stephanie, Trampoline, 21). Unfortunately, repetition did not always seem fruitful, and sometimes it did not help the performer break out of their negative circle.

"If I can't do something, I do it lots of times, and sometimes I get angry with myself." (Rose, Rhythmic gymnastics, 17)

"I was also frustrated because I tried and tried and tried and tried and tried, but it didn't work." (Jane, Ballet and contemporary dance, 16)

**Seek help from others.** Another strategy mentioned was the possibility of asking for help or external advice in order to "unblock" a situation. One strategy, which was not used by all the performers, but which seemed to be useful, was to ask "your coach to change things [or to turn to] other people who have the same [level of] mastery." (Julia, Rhythmic gymnastics, 15)

"And if I can't do it, I'll ask my dance teacher if she has any advice." (Jane, Ballet and contemporary dance, 16)

"In general, I'll ask anyone who's my age and who might be able to include this in their exercises." (Olivia, Rhythmic gymnastics, 15)

They also explained that they observed others to find inspiration and, possibly, a solution to their problem.

"I'm just going to look at her and say to myself, 'Well, maybe she's doing her arms or her feet better or something?' Yeah, and then I'll use that for my benefit." (Ana, Ballet, 20)

"The first thing is that I really listen the other musician. I listen. I say to myself, 'Okay, he's there [on the music sheet]. Right. Okay." (Cloe, Trumpet, 22)

Finally, simply getting some positive feedback sometimes unblocked a situation by comforting performers enough for them to overcome the difficult situation.

"I try to listen to what other people say—because they talk to me too—to try and give myself a bit of confidence." (Olivia, Rhythmic gymnastics, 15)

**Put things into perspective.** Performers also tried to put their situation into perspective or take their minds off their failure. Experience seemed to be key to putting things into perspective and moving on. Athletes who had encountered similar setbacks in the past described drawing on these experiences to reassure themselves that the distress would be temporary.

"I say to myself, 'It's just a bad mood; I know that it's not right now, but it will pass.' So, I deal with it because I've already... every time I've had this, I knew that it would get better afterwards, even if, at the time, it's really... you feel awful, you know that afterwards, it will get better, and then you think about the long term." (Lucy, Ballet, 14)

Similarly, performers highlighted the importance of acknowledging other achievements as a way to mitigate the emotional weight of failure.

"Afterwards, I tell myself that it was good that I managed to do the other difficult things—that I've succeeded—and that helps me to put things into perspective." (Olivia, Rhythmic gymnastics, 15)

Performers also managed to put things into perspective with the help of their "[coach] [...] who always understands" (Emmy, Ballet, 17).

**Externalize.** When a performer is experiencing moments of blockage or difficulty, getting their emotions (e.g., frustration, sadness, anger) out can help them to move on. This process often functioned as a means of emotional release that enabled athletes to "move on" from immediate frustration or distress.

"It's a way I use to move on—laughing with them." (Ian, Diving, 17)

Similarly, few performers recounted moments of crying after training as a natural response to the emotional strain of perceived failure or negative feedback from coaches.

"Yes, so sometimes I'd come home from training and cry because I couldn't do something and because the teacher might have shouted once, and now, yes, that's normal." (Jane, Ballet and contemporary dance, 16)

However, these means of externalising things did not always seem to benefit the performers. Rather, they seemed to be the consequence of negative experiences and reflect an inability to regulate or distance themselves from overwhelming emotional experiences.

> "The fact that I come home after training crying—I think that means I'm not able to take any distance." (Stephanie, Trampoline, 21)

**Stop and connect with your body.** When "nothing goes right", performers stop, take "a break" and "come back to it later" (Maria, Piano, 26). Nathan (Trumpet, 26) even said that he took breaks of several days before getting back to work. As far as dancers and athletes are concerned, the context in which they practised did not allow them to stop so absolutely, but they did pause physically, either to calm their mind and centre themselves on their body once more or to "feel it, visualise it" (Stephanie, Trampoline, 21).

> "I think about it before diving and imagine doing the dive and correcting it—because the coach gives you a report— between two reports, the corrections, and you try to apply what the coach says as much as possible." (Ian, Diving, 17)

### Create a virtuous circle

**The power of common humanity.** In general, common humanity was something that performers seemed to have developed relatively easily. They did not feel alone in their experiences and knew that others had gone through the same things.

> "It helps to tell yourself that it's actually normal and that there's nothing weird or abnormal about simply crying or reacting like that, because other people do it too. [...] So, yes, it's quite nice to realise that, to notice it and to say to yourself, 'It's true, it's normal. You're allowed to react like that, and it's not abnormal.'" (Julia, Rhythmic gymnastics, 15)

Nevertheless, others always seemed to have had an easier time of it: "they can do it, but I can't" (Ryan, Percussion, 18). Performers repeatedly expressed the importance of their peers, whether they were peers in their sporting or artistic activities or peers with no connection to their activity. They were better able to discuss the problems they encountered with peers in their sporting and artistic activities and feel connected to them. The performers sometimes even got help from others to overcome their difficulties or to push themselves forward.

> "So, even though it's true that sometimes you think, 'Yes, comparing yourself to others isn't a good thing,' I think that, on the contrary, it really helps us to progress. And I think that, yes, I would really describe it as positive competition." (Ana, Ballet, 20)

> "If I haven't been able to do something well, I'm going to look at the people who are doing it and ask them for a better way of approaching the move than I have been using." (Rose, Rhythmic gymnastics, 17)

Sharing experiences, whether between peers or between coaches and performers, seemed to be an essential means of learning and of legitimising what the performers were experiencing: "As far as my teacher is concerned [...] she touches me [emotionally] more anyway. I tell myself that she cried too, that she was in the same situation as me, so it's normal. But when it's a student, I think, 'Yeah, but she's not really going through the same thing [as me]." (Emmy, Ballet, 17)

Finally, the very process of taking part in an interview as part of a research project seemed to have been beneficial and cathartic for the performers, highlighting the importance of sharing one's experience to digest it.

> "It's been a great help, and it's also made me realise a lot of things that I hadn't realised before. Whether in relation to my previous schools, experience or whatever." (Linda, Acrobatic rock and roll, 19)

Although performers managed to feel connected to others, they still engaged in activities that put them in competition with them, and this competition sometimes led them to think "that others know more" (Cloe, Trumpet, 22) and "having a feeling of loneliness" (Elisabeth, Viola, 18).

**Space for your inner friend.** Fortunately, even so performers were often quite critical towards themselves, they were also capable of self-care and compassion. They attempted to have caring and constructive inner dialogues: "'This isn't right, Elena. You can't do more than you're capable of, and don't forget why you dance'. And the reason I dance is because I like doing it—I want to—so if I don't want to and I'm not satisfied, I won't get anywhere" (Elena, Ballet and contemporary dance, 17).

Performers tried to motivate themselves by having more caring thoughts: "I say to myself, 'Okay. Now it might not be perfect yet, but it's already a bit better than yesterday'." (Lucy, Ballet, 14). Or, more simply, 'Keep doing that and, in time, it'll get better'. So, in general, I try to encourage myself." (Ryan, Percussion, 18)

However, this caring intrapersonal support often comes with strings attached. Performers do not fully believe in their skills or feel they do not deserve their success: "Well, at that moment, I'm actually very happy and proud to say to myself 'I've succeeded.' Then, on the one hand, I say to myself, 'Was it a fluke?' But then, when I think about it, I say to myself, 'No, that's impossible. I've worked for this, so it can't be a fluke'." (Victoria, Flute, 27)

Performers were able to have a positive inner dialogue and help themselves, but, it seemed, only under certain conditions or if their training was going well.

"I think both. I don't know. I tell myself that something unexpected could always happen. I'm never sure what's going to happen, whether it's on stage or when we're dancing. So, I know I can do it, but if it's better than I thought, sometimes I wonder if I deserve it." (Emmy, Ballet, 17)

These results highlighted the importance of positive feelings and of creating a positive virtuous circle of emotions and thoughts, among other things, by feeling connected and sharing one's experiences with others.

## Acknowledge your social environment

**The importance of support and role models.** Performers stressed the importance of their parents' support: "I know I can count on them if there's a problem at the gym." (Julia, Rhythmic gymnastics, 15). Some, who were away from their families for their training, also emphasised that they felt something was missing.

"It's also hard sometimes, because I think, 'If they were here...'. So, as I left when I was 14, there was a whole part of my childhood when I wasn't with them. Sometimes it's still in the back of my mind. I say to myself, 'But if they were with me, I wouldn't ask myself these questions. I'd carry on, I'd be calmer and more serene'." (Elena, Ballet and contemporary dance, 17)

Parents seemed to be an essential pillar of support, people with whom performers could express their feelings. They sometimes acted as receptacles for the emotion's performers experienced in their artistic and sporting activities: "If I didn't dare respond to something a coach said, or if I was really too angry and didn't know what to do to calm down, well, sometimes we talk about it in the evening to try things out at the next training." (Olivia, Rhythmic gymnastics, 15)

Coaches often seem to have an important place in the athlete's mind. They are role models or guide that they listen to more than other 'voices'. They sometimes put their coach on "a bit of a pedestal" (Cloe, Trumpet, 22). Coaches' experiences even seem to have more value than their own:

"For my teacher, I think it's different when she tells me that because I think it's also maybe due to experience or something, but I don't know. She touches me more, in any case." (Emmy, Ballet, 17)

Performers value their experience enormously, so they feel they "must trust [them and] follow [them] with [their] eyes closed" (Ian, Diving, 17).

**Understand each other's reality.** The performers wished that their coaches understood them better. Their coaches' validation of their experiences seemed crucial to them. In the way coaches communicate, performers sometimes have the impression that they don't understand their reality. Coaches don't "necessarily put themselves in [their] place" (Elisabeth, Viola, 18). Performers expressed a strong need for their coaches to recognize their efforts and validate their subjective experiences, especially during challenging moments.

> "And often, at times like that, the coaches will say to you, 'That was a stupid mistake. Concentrate!' when you already have the impression that you're concentrating, that you're going all out." (Olivia, Rhythmic gymnastics, 15)

> "For example, last year we had to learn a new movement. I didn't even know what it was, so when he yells at me like that and tells you to do it like that and you can't do it and you're tense. It's horrible." (Isabel, Volleyball, 16)

Conversely, when coaches demonstrated an awareness of the performer's individuality and adapted their approach accordingly, this was perceived as deeply supportive. Performers want coaches to adapt and understand who they are.

> "Well, I have the impression that he understood the person I was, at last, and so he adapted well. He knew that he didn't need to be harsh or to shout at me—because he could be harsh with others. I've heard him be harsh with others. But he understood that with me, it wasn't any use." (Maria, Piano, 26)

**The power of communication.** Performers emphasised the importance of receiving feedback. The majority of performers explained that their coach mainly gave "technical corrections [but] not too many motivational phrases" (Rose, Rhythmic gymnastics, 17). Positive, motivational feedback doesn't seem to be the norm, but performers express the need to hear it.

> "When I'm in a dance class and I'm not doing well, because of the dancing, I'd really need the teacher to say something to me, to compliment me or just say to me, 'Ah! This time you've done better than the other days.' And I think that's what really cheers me up." (Lucy, Ballet, 14)"

> It's very rare when he tells me that things are going really well, but when they're not, he often tells me"(Ryan, Percussion, 18).

> "First of all, it's nicer for everyone if it's positive and the coach is nice, but I also think it's going to bring better results, because if you go to a training session or a lesson and you're stressed, you're afraid to see the teacher, or you're just afraid of getting remarks, well, you'll be more tense; it's not going to go as well." (Emmy, Ballet, 17)

Fortunately, some explain that "[coaches] are changing, and they want to correct [them] in a positive way" (Linda, Acrobatic rock and roll, 19).

> "He's respectful and when he sees that things aren't going well, he knows and tries to calm things down. I like it because he asks you what you want to do and also, last year, [...] we were doing technique and we were really doing what I couldn't do. So he'd give me advice and even if I couldn't do it well, he'd make me do the exercise again until I got it right, at least once or twice. From time to time, I think that's good, it's encouraging." (Isabel, Volleyball, 16)

However, performers also pointed out that negative feedback and corrections were sometimes more informative than positive feedback.

> "They have to make you progress and they also have to be a bit hard on you. They also have to give you a lot of correction and they have to say if there's something that's not working." (Jane, Ballet and contemporary dance, 16)

Performance activities also seemed to create a hierarchy between the performer and the coach. Coaches "know better than [the performer]" (Maria, Piano, 26). This hierarchy and distance lead performers to not "really know how to ask questions" (Lucy, Ballet, 14).

But, performers highlighted the importance of sharing their needs: "If I need to talk about something with my coach, I know—he's always told me that he'll always be very open, and it's very nice to know that if things aren't going well—there's always someone to talk to. And the coach, I think, is the first person I'd talk to if there were a problem with the sport." (Ian, Diving, 17)

Seeking validation and attention from their coaches seems very important. They seemed to need the commitment and recognition of their coach.

> "It's harder when you don't get any corrections during the lesson or when the teacher doesn't look at you at all." (Elena, Ballet and contemporary dance, 17)

It seems that the absence of comments from coaches is interpreted as a failure or understood by some performers as a negative judgement of their performance.

### Learn about yourself

**Gaining more experience.**  Participants were asked what they would need to be more successful in overcoming difficult situations and developing self-compassion. It seemed that performers needed to live through their "bad experiences" (Lucy, Ballet, 14) and learn from them to develop coping skills. However, gaining experience could also involve "exchanges between different students" (Maria, Piano, 26).

They also expressed the need for their more experienced coaches to share their experiences and sometimes validate their feelings or their journey without wanting them to be totally performance-oriented.

Finally, it seems important to remain open to the world to gain experience: "Read articles on the subject because I think I might find something that will ultimately help me" (Sara, Artistic gymnastics, 19).

**Get help from specialists.**  They also suggested being "accompanied [by] the [coach], but also by more experienced [performers]" (Ryan, Percussion, 18). It seems important for them to be monitored 'physically and psychologically' (Stephanie, Trampoline, 21). Some of them already have the support of mental coaches and it helps them a lot:

> 'I already like the fact that [the mental coach] comes to our group every other week for half an hour to talk. We talk about the competitions we've been to, we plan our goals for the next ones, and we talk a bit about how things are going in training. (Olivia, Rhythmic gymnastics, 15)

> "I think support is the most ideal thing, because you can open up about yourself and the person in front of you is there to help you anyway." (Sara, Artistic gymnastics, 19)

From what they say, it seems important that help comes "from the outside, because [...] sometimes [they're] too wrapped up in [their] practice, and [it] prevents [them] from being objective" (Cloe, Trumpet, 22). Depending on the performer, "[they] have to learn [how to become compassionate], yes. There are some people who do it very well on their own and others who might need help to do it" (Felix, Badminton, 18).

## Discussion

The present study aimed to examine the concept of self-compassion among young performers—athletes, dancers and musicians—to better understand how it might or might not be integrated into their training and practice when

facing difficulties. In addition, we were interested in the social factors that might have an impact on whether or not self-compassion develops.

The participants found the concept of self-compassion particularly interesting. Although some studies have highlighted athletes' reticence and fears regarding self-compassion [51], this was not the case among our performers, even if they did emphasize the need for self-criticism. The present results corroborated those of [30], who found that athletes relied on self-criticism in sporting contexts. The performers in this study explained that self-criticism enabled them to move forward and push themselves—results echoed by [29], who found that performers relied on self-criticism to improve. [28] even highlighted that the world of performance "normalize criticism and self-criticism to the point where athletes and coaches often promote and foster the utility of criticism and self-criticism as a way to improve, even though athletes identify that they did not fully believe that self-criticism was the best approach" (28 p272). The results of our study emphasize the usefulness of self-criticism, but also its harmful effects. Therefore, counterbalancing self-criticism by learning to be self-compassionate could be a way of toning down your inner critic without eliminating it altogether.

Indeed, some of the performers in our study had reservations about the usefulness of this constant self-criticism. As a matter of fact, [52] highlighted the risks that self-criticism represented for achieving one's goals. Furthermore, [7] also showed that self-criticism was negatively linked to post-competition performance evaluations. It would, therefore, be interesting to help performers develop a more constructive inner critical dialogue. The idea is not to suppress our inner criticism but to nuance it or compensate for it by having a more benevolent inner dialogue [9]. Finding a balance between one's critical self and one's benevolent self may involve being aware of what one is truly capable of achieving, being self-confident [8], and being as close as possible to that achievement. However, some of the performers in this study seemed to have set their expectations far too high. Most of them were striving for perfection and were intolerant of mistakes. [53] revealed, for example, the necessity, for professional musicians, to develop a balance between striving to meet their own standards and the risk of succumbing to self-pity during periods of failure. These perfection-seeking tendencies, coupled with the idea that nothing is ever good enough, could contribute to their harsh and negative self-criticisms [12]. If performers learned to value their actions through self-compassion, their perfectionist tendencies would likely diminish [54] and their motivation would increase [55]. However, performers could easily offer support to peers, but they struggled to extend the same kindness to themselves, highlighting a self-compassion gap. Indeed, [56] showed a higher level of apprehension towards self-compassion than towards compassion for others in a sample of UK athletes.

There is also a question about whether our performers were too focused on their results rather than on the processes of learning, practising and training, on the task they must accomplish to perform [57]. It seemed that the more performers dwelt on the results they wanted to achieve, the harder it was for them to distance themselves and be in the present moment (the mindfulness component of self-compassion), and the more they fell into a negative vicious circle and over-identified, cognitively and emotionally, with the situation they were living. The present results, and those of other researchers [38,58], showed that performers had difficulty being mindful when faced with setbacks, and they were often caught up in their thoughts, which were frequently oriented towards self-criticism. They also found it difficult to detach themselves from their emotions, including the anger, frustration, shame or sadness they expressed at not having achieved something. In line with our findings, [23] highlighted that over-identification among musicians could lead to performance anxiety. Learning to develop other ways of being true to oneself, therefore, seems equally relevant for musicians, dancers and athletes. Performers realised that positive things led to more positive things. Of course, being always positive and supportive is impossible, however it seems that the earlier a virtuous circle is established (self-kindness, positive emotions and well-being), the more easily and quickly performers will be able to get out of difficult situations. [32] underlined that athletes with higher scores of self-compassion had higher levels of goal re-engagement, less rumination and less negative emotions.

The question is how can performers overcome difficult situations without being over-critical or over-identify themselves to the situation? Our performers exposed different strategies they used to get out of difficult situations they might

encounter during their practice (e.g., difficulty executing a movement or repeated mistakes). They try to return to a basic movement or break the situation down into smaller elements to understand what's happening. As highlighted by [59] focusing on the task increase the feeling of control in what they are doing. Therefore, focusing on easier movements or actions to avoid making the same mistake over and over again seems to be a way of avoiding a cognitive or emotional whirlwind. It helps them to calm down and not get carried away by the difficulty they are encountering. Unfortunately, even if this technique is sometimes useful, the performers still don't always succeed at the end and often get carried away by their frustration. This is when it's a good time to show some self-compassion and possibly turn to other strategies. Another strategy used, which not all performers agreed on, was to turn to others for help (e.g., coach or peers). They ask their coach for technical help, for example, or also seek comfort and support from their peers. Asking for help sometimes enabled them not to get carried away by their internal criticism and be aware of their strengths as studied by [60]. The performers in our study tried to put the situation into perspective and play it down, nevertheless they expressed how this did not always work and that their self-critical internal dialogue often took over. They also explained that it was easier for them to put things into perspective after the event rather than when it happened. Another strategy was to externalise what was going on the inside (i.e., emotions). This was done, for example, by crying or laughing with their peers in order to play it down. Nevertheless, the performers highlighted that crying was perhaps the result of over-identification with the situation or with negative internal cognitions. They also tried to refocus on something else, such as going back to a basic movement that they mastered to try to feel a certain physical sensation. Their aim was to stay focused on what they were doing and overcome their difficulties without getting carried away by their emotions and starting to ruminate.

These strategies may sometimes help in the moment, however, as mentioned earlier, maintaining a positive virtuous circles, through cognition and emotions, may have a powerful impact during times of setbacks. To create and maintain this virtuous circle, it seems that sharing their experiences with others is fundamental. [41] highlighted that coaches and teammates sharing information and conditional supports can increase level of self-compassion. The performers in this study realised that they were not alone in having difficulties. This shared humanity helped them to overcome difficult situations. They also sought comfort or help from others, but, as our findings revealed, it was interesting to note how performers minimised or invalidated others' positive discourses and kind words toward themselves. It would seem important, therefore, to help performers learn to accept what others have to say about them so that they can subsequently accept saying it to themselves. This learning process cannot be done alone, which is why it is so important to have the support of others (coaches, dance instructors, music teachers or parents) [41]. The environments in which the performers in this study practised did not always seem to encourage sharing with others, particularly with coaches. Also, [61] have highlighted that emotions, thoughts, and reactions to negative or emotionally difficult scenarios were lighter when athletes presented a higher level of self-compassion.

As mentioned above, adequate social support can strengthen our ability to be kind to ourselves [38]. Some researchers have shown that recognising and expressing one's emotions in a healthy way can foster compassion towards ourselves [6]. This may involve opening up to our emotions, sharing them with people we trust or writing in a diary. According to our study, parents often fulfilled the role of unconditional supporter and attentive listener, somebody who could accept their children's moods or listen to what had not worked during training. [37] highlighted the importance of parents in the development of self-compassion. Coaches play the role of guides that should enable performers to assess themselves accurately—they are not there to condemn each mistake or setback encountered [38,40]. Moreover, we know that young performers admire their coaches and take them as role models [10,37]. However, it is also crucial that young performers learn to validate their own internal judgement. Indeed, in our study, performers are all at an advanced stage in their career where they possess a level of expertise that deserves to be recognised and integrated into their learning and developmental processes. Encouraging self-compassion in performers and enabling them to validate their own judgement can be facilitated by establishing a climate of listening and open exchange. The hierarchical systems found in the world of performance, especially in the coach-performer relationship, sometimes make it difficult for them to confide in each other or

express the difficulties they are experiencing. However, sharing experiences—all the more so with people who have some expertise in their field—would help performers develop new skills to deal with their difficulties [62]. In addition, impactful relationships and legitimising what we have experienced can help us better accept the difficulties we encounter so that we can better respond to them [11]. The performers in this study also expressed the need for their coach to understand their reality—that coaches should sometimes be able to put themselves in their shoes. It seemed important, therefore, to create a space within which the coach could learn to understand the performer's limits and thus be able to adapt their discourse to the performer's reality.

Finally, performers talked about the skills needed to manage difficult moments—when a performer is struggling to break out of a negative loop during which they are unable to achieve something. It seems that these skills were developed individually and with experience. This suggests that the development of self-compassion may occur only in an unstructured and implicit manner through experience, implying that performers can't deliberately learn or strengthen these skills. However, various authors have argued that developing self-compassion skills through guided support is possible [3] and makes it even possible to recover more quickly from failure and persevere in the face of difficulty [63]. When asking what they would need to develop self-compassion, performers told they needed help from a specialist or their coach and think that through experience and knowing yourself, you could develop a more compassionate self. Indeed, it seems important that performers understand their patterns and behaviours (e.g., negative circle of self-criticism) before they can welcome help or hear and accept the experiences of others as a guide to developing self-compassion.

The present study had some limitations. There was inevitably a selection bias, given that the performers participated voluntarily. They were already interested in the study's theme in one way or another and wanted to find out more about themselves. Moreover, the predominance of female participants may have influenced the findings, highlighting the need for future research with a more gender-balanced sample to explore potential gender-related differences in experience and perspective. Participants were also selected at random when it might have been interesting to pre-select participants according to their level of self-compassion to compare those who seemed to be naturally compassionate against those who were not. While the present study's results enabled us to refine our understanding of the concept of self-compassion among performers, a study with a larger cohort over a longer period, possibly combined with quantitative results, would enable us to examine this theme in greater depth and provide more support for our findings.

## Conclusion, future research and practical implications

This study provides an understanding of how self-compassion is experienced and developed among young high-level performers—athletes, dancers, and musicians—when facing setbacks. Although, many performers showed interest in the concept of self-compassion, it remains relatively unknown within performance domains. One positive side effect of this research is the information transmitted to performers. The discovery of this concept opened up new possibilities for them in terms of how they might react to difficult situations and failures. Being aware of one's emotions, feelings, thoughts and sensations, without judging them, can help us to accept our difficulties and face them with self-compassion. While the performers valued the development of self-compassion, they also acknowledged the necessity of self-criticism, even if it had an emotional cost. These insights reinforce the need for performers to cultivate a more constructive and balanced inner dialogue, where self-kindness can coexist with self-criticism. However, self-compassion does not appear to be an innate skill, but it can be learned and developed with the appropriate support and guidance. It would seem important, therefore, to prepare training courses or pass on the necessary information so that performers and their coaches understand how to approach this concept and develop it. As part of their mental preparation, performers could be taught about self-compassion—something which could accompany them toward well-being and success in their performances. Professionals can help performers and their coaches develop a healthier relationship with themselves to reach their full potential. Finally, this study highlights the importance of addressing the cultural norms and hierarchies within different performance disciplines, which can influence how self-compassion is received and practiced. Future research should explore how

these norms shape performers' experiences and extend the inquiry to coaches, whose perspectives and practices are central to shaping compassionate performance environments.

## Supporting information

**S1 Table. Interview guide.** It is the interview guide used for the 27 semi-structured interviews.
(DOCX)

**S2 Table. Sentences representing the three components of self-compassion.** These are the three sentences presented to the participants during the interviews.
(DOCX)

**S1 Checklist. Checklist: Research on and with children and adolescents under the age of 18.** This is the SwissEthics checklist that regulates research with children and adolescents in Switzerland.
(PDF)

## Acknowledgments

We sincerely thank all the participants who gave up their valuable time to share their stories with us and Darren Hart, of *Publish or Perish*, for proofreading this article.

## Author contributions

**Conceptualization:** Céline Kosirnik.

**Formal analysis:** Céline Kosirnik.

**Investigation:** Céline Kosirnik.

**Methodology:** Céline Kosirnik.

**Supervision:** Roberta Antonini Philippe, Valentino Pomini.

**Writing – original draft:** Céline Kosirnik.

**Writing – review & editing:** Céline Kosirnik, Roberta Antonini Philippe.

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
