## [Decision Letter · Decision Letter 0]

May 19 2025

PONE-D-24-51767Navigating pathways to develop self-compassion in sport, dance, and music: the pivotal influence of the social environmentPLOS ONE

Dear Dr. Kosirnik,

Thank you for submitting your manuscript to PLOS ONE. After careful consideration, we feel that it has merit but does not fully meet PLOS ONE’s publication criteria as it currently stands. Therefore, we invite you to submit a revised version of the manuscript that addresses the points raised during the review process.

We look forward to receiving your revised manuscript.

Kind regards,

Mustafa Can KOC, PhD

Academic Editor

PLOS ONE

Additional Editor Comments:

Reviewer-1.

Overall: This is an excellent manuscript, and I thank the authors for conducting this important piece of research which is a valuable contribution to the social science/psychology literature base. Here are some minor corrections I have listed -

Generally: Make sure your citations are consistent across the whole manuscript and that they are in line with the journal criteria.

Line 137: Try not to just list citations like this here, explain directly what points they talked about, and how your works builds on them.

Line 140: Maybe insert ‘study’ after ‘qualitative research’

Line 164: Can you be more specific about the type of qualitative study this was? Did you adopt a specific research design?

Line 177: I’m not sure what a resource person is, this may confuse readers. Can you think of a better term?

Line 197: Can you provide a citation that guided your overarching approach to analysis? I know you provide some for the coding, but what about the ‘mixed inductive-deductive analysis’ more generally.

Line 323: I’m not sure I like the term ‘Get back to simple mechanism’. Maybe replace it with something catchier, like ‘Back to the basics’?

Line 371: Can you provide some more descriptive commentary on this theme here, outlining in a bit more detail what this process involved and why it is significant (alongside the data you have provided).

Line 387: Same as point above, please provide a little more descriptive commentary.

Line 524: Expand a little more why this validation was crucial to them.

Line 649: On the published version of this paper I examined, this is contained on page 272, not 18. Please double check this and make sure it is correct.

Line 782: I don’t understand how ‘training or learning to manage these situations’ with compassion is ‘not necessarily possible’. Can you please provide more explanation behind this point or clarify what this means in your response to me.

Reviewer-2.

This paper addresses a highly pertinent topic —self-compassion in performance domains such as sports, dance, and music. The qualitative insights provided here underscore the potential benefits of self-compassion for athletes, dancers, and musicians, as well as the importance of a supportive social environment in nurturing these attitudes and coping strategies. The study has clear relevance, given the growing interest in mental well-being and resilience within high-performance settings. It adds to the relatively sparse literature on self-compassion among pre-professional performers, thereby filling an important gap.

General Comment

Despite these merits, the manuscript would benefit from the inclusion of more visually engaging data presentations (e.g., thematic maps, diagrams, tables). Currently, the results rely heavily on textual explanation, which can impede a clearer interpretation of the findings and limit readers’ ability to construct a cohesive narrative of the data.

Specific Points for Improvement

Title Specificity: The current title does not reflect that participants were volunteers drawn from a convenience sample. Specifying this aspect would clarify the study’s scope and methodological constraints.

Gender Distribution: The sample is heavily female-dominated, but the paper lacks a thorough discussion of how this imbalance might affect the findings. Addressing potential gender-related biases or limitations would strengthen the study’s credibility.

Validation of Interview Guide: The methodological section does not provide robust detail on how the interview instrument was validated (e.g., pilot testing, external expert reviews). Including this information would help readers assess the rigour of the qualitative approach.

Depth of Recommendations: While recommendations on how to foster self-compassion are offered, they remain overly general. Providing more detailed, step-by-step or evidence-informed guidelines for coaches, trainers, or educators would greatly enhance the paper’s applicability, with proper referencing.

Conciseness: There is repetition in both the introduction and the discussion, particularly around conceptual definitions. A more succinct treatment of core concepts would improve readability and allow for deeper engagement with new insights.

Conclusion: The conclusion is coherent but somewhat broad. Highlighting a clear, specific contribution —such as identifying unique themes or actionable strategies for different types of performers— would demonstrate the paper’s added value.

Overall, this manuscript covers an essential domain of performance psychology and offers thoughtful perspectives on the development of self-compassion. With more focused revisions and the inclusion of compelling data visualizations, it has the potential to make a substantial contribution to the literature on mental well-being in sport, dance, and music contexts.

Reviewers' comments:

Reviewer's Responses to Questions

**Comments to the Author**

1. Is the manuscript technically sound, and do the data support the conclusions?

Reviewer #1: Yes

Reviewer #2: Yes

2. Has the statistical analysis been performed appropriately and rigorously? 

Reviewer #1: N/A

Reviewer #2: N/A

3. Have the authors made all data underlying the findings in their manuscript fully available?

Reviewer #1: No

Reviewer #2: No

4. Is the manuscript presented in an intelligible fashion and written in standard English?

Reviewer #1: Yes

Reviewer #2: Yes

5. Review Comments to the Author

Reviewer #1: Overall: This is an excellent manuscript, and I thank the authors for conducting this important piece of research which is a valuable contribution to the social science/psychology literature base. Here are some minor corrections I have listed -

Generally: Make sure your citations are consistent across the whole manuscript and that they are in line with the journal criteria.

Line 137: Try not to just list citations like this here, explain directly what points they talked about, and how your works builds on them.

Line 140: Maybe insert ‘study’ after ‘qualitative research’

Line 164: Can you be more specific about the type of qualitative study this was? Did you adopt a specific research design?

Line 177: I’m not sure what a resource person is, this may confuse readers. Can you think of a better term?

Line 197: Can you provide a citation that guided your overarching approach to analysis? I know you provide some for the coding, but what about the ‘mixed inductive-deductive analysis’ more generally.

Line 323: I’m not sure I like the term ‘Get back to simple mechanism’. Maybe replace it with something catchier, like ‘Back to the basics’?

Line 371: Can you provide some more descriptive commentary on this theme here, outlining in a bit more detail what this process involved and why it is significant (alongside the data you have provided).

Line 387: Same as point above, please provide a little more descriptive commentary.

Line 524: Expand a little more why this validation was crucial to them.

Line 649: On the published version of this paper I examined, this is contained on page 272, not 18. Please double check this and make sure it is correct.

Line 782: I don’t understand how ‘training or learning to manage these situations’ with compassion is ‘not necessarily possible’. Can you please provide more explanation behind this point or clarify what this means in your response to me.

Reviewer #2: This paper addresses a highly pertinent topic —self-compassion in performance domains such as sports, dance, and music. The qualitative insights provided here underscore the potential benefits of self-compassion for athletes, dancers, and musicians, as well as the importance of a supportive social environment in nurturing these attitudes and coping strategies. The study has clear relevance, given the growing interest in mental well-being and resilience within high-performance settings. It adds to the relatively sparse literature on self-compassion among pre-professional performers, thereby filling an important gap.

General Comment

Despite these merits, the manuscript would benefit from the inclusion of more visually engaging data presentations (e.g., thematic maps, diagrams, tables). Currently, the results rely heavily on textual explanation, which can impede a clearer interpretation of the findings and limit readers’ ability to construct a cohesive narrative of the data.

Specific Points for Improvement

Title Specificity: The current title does not reflect that participants were volunteers drawn from a convenience sample. Specifying this aspect would clarify the study’s scope and methodological constraints.

Gender Distribution: The sample is heavily female-dominated, but the paper lacks a thorough discussion of how this imbalance might affect the findings. Addressing potential gender-related biases or limitations would strengthen the study’s credibility.

Validation of Interview Guide: The methodological section does not provide robust detail on how the interview instrument was validated (e.g., pilot testing, external expert reviews). Including this information would help readers assess the rigour of the qualitative approach.

Depth of Recommendations: While recommendations on how to foster self-compassion are offered, they remain overly general. Providing more detailed, step-by-step or evidence-informed guidelines for coaches, trainers, or educators would greatly enhance the paper’s applicability, with proper referencing.

Conciseness: There is repetition in both the introduction and the discussion, particularly around conceptual definitions. A more succinct treatment of core concepts would improve readability and allow for deeper engagement with new insights.

Conclusion: The conclusion is coherent but somewhat broad. Highlighting a clear, specific contribution —such as identifying unique themes or actionable strategies for different types of performers— would demonstrate the paper’s added value.

Overall, this manuscript covers an essential domain of performance psychology and offers thoughtful perspectives on the development of self-compassion. With more focused revisions and the inclusion of compelling data visualizations, it has the potential to make a substantial contribution to the literature on mental well-being in sport, dance, and music contexts.

6. PLOS authors have the option to publish the peer review history of their article (what does this mean? ). If published, this will include your full peer review and any attached files.

**Do you want your identity to be public for this peer review?** For information about this choice, including consent withdrawal, please see our Privacy Policy .

Reviewer #1: **Yes: ** Anthony J Bell

Reviewer #2: **Yes: ** Gabriel Alves

---

## [Author Response · Author response to Decision Letter 1]

31 May 2025

You can find, below, my responses to the reviewers’ comments. I hope they answer to their wish. Also, I've created a new Table (as asked from a reviewer), and I've modified the Figure 1. The new table was added in the new manuscript and the modified figure was re submitted.

Reviewer-1.

Overall: This is an excellent manuscript, and I thank the authors for conducting this important piece of research which is a valuable contribution to the social-science/psychology literature base. Here are some minor corrections I have listed -

• Thank you very much for your comment. Hopefully, our work will help change things in the performance world too.

Generally: Make sure your citations are consistent across the whole manuscript and that they are in line with the journal criteria.

• I've checked all the citations and normally everything's clear. I've added the doi, where possible.

Line 137: Try not to just list citations like this here, explain directly what points they talked about, and how your works builds on them.

• I've added information about the research presented to have a clearer view of the important points regarding their study.

Line 140: Maybe insert ‘study’ after ‘qualitative research’

• The word study has been added.

Line 164: Can you be more specific about the type of qualitative study this was? Did you adopt a specific research design?

• I explain that it's a one-on-one, semi-structured interview. I've added that it's qualitative research allowing us to understand and explore the concept of self-compassion in a flexible yet guided way to ensure reproductability.

Line 177: I’m not sure what a resource person is, this may confuse readers. Can you think of a better term?

• I've changed the word resource by expert of the discipline. We did a pilot testing with an expert of the field dance, music and sport to ensure that the questions were comprehensive.

Line 197: Can you provide a citation that guided your overarching approach to analysis? I know you provide some for the coding, but what about the ‘mixed inductive-deductive analysis’ more generally.

• I've added citations that support this analytical process.

Line 323: I’m not sure I like the term ‘Get back to simple mechanism’. Maybe replace it with something catchier, like ‘Back to the basics’?

• Great suggestion! I've changed the title to "Back to the basics"

Line 371: Can you provide some more descriptive commentary on this theme here, outlining in a bit more detail what this process involved and why it is significant (alongside the data you have provided).

• I've added few commentaries.

Line 387: Same as point above, please provide a little more descriptive commentary.

• I've added few commentaries.

Line 524: Expand a little more why this validation was crucial to them.

• I've added few commentaries.

Line 649: On the published version of this paper I examined, this is contained on page 272, not 18. Please double check this and make sure it is correct.

• Indeed, I'm sorry for this mistake. It has been changed.

Line 782: I don’t understand how ‘training or learning to manage these situations’ with compassion is ‘not necessarily possible’. Can you please provide more explanation behind this point or clarify what this means in your response to me.

• I've tried to make it clearer.

Reviewer-2.

This paper addresses a highly pertinent topic —self-compassion in performance domains such as sports, dance, and music. The qualitative insights provided here underscore the potential benefits of self-compassion for athletes, dancers, and musicians, as well as the importance of a supportive social environment in nurturing these attitudes and coping strategies. The study has clear relevance, given the growing interest in mental well-being and resilience within high- performance settings. It adds to the relatively sparse literature on self-compassion among pre-professional performers, thereby filling an important gap.

• Thank you very much for your positive comment.

General Comment

Despite these merits, the manuscript would benefit from the inclusion of more visually engaging data presentations (e.g., thematic maps, diagrams, tables). Currently, the results rely heavily on textual explanation, which can impede a clearer interpretation of the findings and limit readers’ ability to construct a cohesive narrative of the data.

• There is one figure resuming the data and I've added a table exposing the themes and subthemes as well as the results overview in one sentence. Is it clearer like this?

Specific Points for Improvement

Title Specificity: The current title does not reflect that participants were volunteers drawn from a convenience sample. Specifying this aspect would clarify the study’s scope and methodological constraints.

• I've modified the title for the following: Navigating pathways to develop self-compassion in sport, dance, and music: qualitative insights from volunteer participants on balancing criticism and compassion, the pivotal influence of the social environment and strategies to overcome setbacks. I hope it clarifies the scope and methodological constraints, as suggested.

Gender Distribution: The sample is heavily female-dominated, but the paper lacks a thorough discussion of how this imbalance might affect the findings. Addressing potential gender-related biases or limitations would strengthen the study’s credibility.

• I've added this point in the limitations.

Validation of Interview Guide: The methodological section does not provide robust detail on how the interview instrument was validated (e.g., pilot testing, external expert reviews). Including this information would help readers assess the rigour of the qualitative approach.

• One sentence was added in the section "Semi-structured interviews", explaining that the interview guide was discussed and tested with an expert of each discipline (dance, music and one sport).

Depth of Recommendations: While recommendations on how to foster self-compassion are offered, they remain overly general. Providing more detailed, step-by-step or evidence-informed guidelines for coaches, trainers, or educators would greatly enhance the paper’s applicability, with proper referencing.

• Given that the findings of the present study concern exclusively performers, we find it essential to also incorporate the perspectives of coaches to develop more detailed and applicable recommendations. A complementary article is currently in preparation, focusing on coaches as the target population, with the objective of formulating clearer, evidence-based recommendations tailored to their needs, in dialogue with the insights presented in this study.

Conciseness: There is repetition in both the introduction and the discussion, particularly around conceptual definitions. A more succinct treatment of core concepts would improve readability and allow for deeper engagement with new insights.

• I've tried to erase repetitions and combine paragraphs that could be combined.

Conclusion: The conclusion is coherent but somewhat broad. Highlighting a clear, specific contribution —such as identifying unique themes or actionable strategies for different types of performers— would demonstrate the paper’s added value.

• I tried to identify more clearly the unique themes of this study and propose strategies to develop self-compassion. I hope it answers your wish for a more specific conclusion.

Overall, this manuscript covers an essential domain of performance psychology and offers thoughtful perspectives on the development of self-compassion. With more focused revisions and the inclusion of compelling data visualizations, it has the potential to make a substantial contribution to the literature on mental well-being in sport, dance, and music contexts.

---

## [Editor Report · Decision Letter 1]

Navigating pathways to develop self-compassion in sport, dance, and music: qualitative insights from volunteer participants on balancing criticism and compassion, the pivotal influence of the social environment and strategies to overcome setbacks

PONE-D-24-51767R1

Dear Kosirnik,

We’re pleased to inform you that your manuscript has been judged scientifically suitable for publication and will be formally accepted for publication once it meets all outstanding technical requirements.

Kind regards,

Mustafa Can KOC, PhD

Academic Editor

PLOS ONE
---

## [Editor Report · Acceptance letter]

PONE-D-24-51767R1

PLOS ONE

Dear Dr. Kosirnik,

I'm pleased to inform you that your manuscript has been deemed suitable for publication in PLOS ONE. Congratulations! Your manuscript is now being handed over to our production team.

Kind regards,

on behalf of

Assoc.Prof. Mustafa Can KOC

Academic Editor

PLOS ONE